# Effects of Deformation at High, Medium, and Cryogenic Temperatures on the Microstructures and Mechanical Properties of Al-Zn-Mg-Cu Alloys

**DOI:** 10.3390/ma15196955

**Published:** 2022-10-07

**Authors:** Wenxue Zhang, Youping Yi, Shiquan Huang, Hailin He, Fei Dong

**Affiliations:** 1Research Institute of Light Alloy, Central South University, Changsha 410083, China; 2State Key Laboratory of High Performance Complex Manufacturing, Changsha 410083, China; 3School of Mechanical and Electrical Engineering, Central South University, Changsha 410083, China

**Keywords:** Al-Zn-Mg-Cu alloy, deformation temperature, microstructures, mechanical properties

## Abstract

Thermomechanical treatment is an effective way to refine the coarse microstructures of aluminum alloys. In this work, multiaxial forging deformation at high, medium, and cryogenic temperatures (i.e., 450, 250, and −180 °C, respectively) was performed on 7085 Al-Zn-Mg-Cu alloys, and its effect on the microstructure evolution and mechanical properties during the subsequent T6 heat treatment process was studied. The results revealed that the coarse particles were broken into finer particles when deformed at cryogenic temperatures, promoting the dissolution of the material after solid solution treatment. Dynamic recrystallization occurred when deformed at 450 °C; however, more dislocations and substructures were found in the samples deformed at 250 and −180 °C, causing static recrystallization after solid solution treatment. The cryogenic deformed sample exhibited a more intense and homogeneous precipitation phase distribution. The strength of the sample deformed at high temperature was high, but its elongation was low, while the strength of the sample deformed at medium temperature was low. The microstructure refinement of the cryogenic deformed sample led to high comprehensive mechanical properties, with an ultimate tensile strength of 535 MPa, a yield strength of 506 MPa, and a fracture elongation of 11.1%.

## 1. Introduction

High-strength aluminum alloys have been widely used to fabricate large transition rings in carrier rockets due to their high specific strength, high specific stiffness, and outstanding mechanical properties at high and low temperatures [1,2,3,4,5]. Compared to a conventional Al-Cu-Mn series alloy, the Al-Zn-Mg-Cu series alloy exhibits higher strength as a potential material for large transition rings.

However, coarse second-phase particles are easily formed in the Al-Zn-Mg-Cu alloy during the casting process due to its high alloying degree [6,7], especially for large ingots with lower cooling rates. The coarse particles consume large amounts of Zn and Mg, causing a precipitation-free zone (PFZ) to form around the particles. The coarse particles also act as nucleation sites for cracks, which leads to their poor plasticity. In addition, inhomogeneous and fiber grain structures usually occur during the conventional hot deformation process, which can induce high anisotropic mechanical properties [8,9].

Severe plastic deformation (SPD) is an effective technique used to refine the microstructures of Al-Zn-Mg-Cu alloys [10]. Wu et al. [11] fabricated a nanolaminate 7085 alloy with an average thickness of 27 nm by surface mechanical grinding, and the hardness increased to 2.86 GPa. Ebrahimi et al. [12] also reported the formation of an ultrafine grain structure during equal channel angular extrusion, which led to the increased strength and hardness of the Al-Zn-Mg-Cu alloy. Zhang et al. [13] significantly increased the mechanical properties of the 7085 alloy through cryo-rolling and peak aging treatment. The yield strength (YS), ultimate tensile strength (UTS), and elongation (El) of the 7085 alloys increased to 635 MPa, 674 MPa, and 11.0%, respectively. Although the strength of the Al-Zn-Mg-Cu alloys increased after SPD, their ductility usually decreased. Mei et al. [14] investigated the effect of rolling deformation on the mechanical properties of the Al-Zn-Mg-Cu alloy and found that the alloy exhibited a better combination of strength and ductility when processed by cryo-rolling and warm rolling.

Many researchers have attempted to regulate the precipitates and enhance the mechanical properties of the Al-Zn-Mg-Cu alloys, with thermomechanical treatment being an effective approach for aluminum alloys. Liu et al. [15] discovered that the residual stress in the alloy promoted the nucleation of precipitates for the duration of the artificial aging, which led to the formation of the finely dispersed GP II region and η’ precipitates in the 7085 alloy. Zhang et al. [16] found that many recrystallized grains and subgrains had formed in the 7055 alloy during the hot deformation process, which altered the precipitation behavior of the η, T, S, and Y phases. Xiang et al. [17] reported that homogenization followed by warm deformation accelerated the precipitation of the Al_3_Zr particles and suppressed dynamic recrystallization. Consequently, the fibrous microstructures were maintained and the strength of the alloy was improved.

Although there are many studies on the grain structures and precipitated phase of the Al-Zn-Mg-Cu alloy, there is little research on the influence of deformation parameters on the refinement of second-phase particles and the dissolution behaviors of coarse particles in the heat treatment process. Regarding the grain structure, most studies focus on dynamic recrystallization behaviors during the hot deformation process, while the influence of static recrystallization on the strength and plasticity of the material has not been reported previously. In this study, the effects of high, medium, and cryogenic temperatures (i.e., 450, 250, and −180 °C, respectively) in forging deformation on the second-phase particles, grain structures, and mechanical properties of the 7085 Al-Zn-Mg-Cu alloy were studied. The mechanisms for grain refinement and improvements in mechanical properties were discussed.

## 2. Materials and Methods

The initial sample was a hot-forged 7085 Al-Zn-Mg-Cu alloy, with a nominal chemical composition of (wt%) 7.5 Zn, 1.5 Mg, 1.4 Cu, 0.15 Zr, and balance aluminum. Three hyperrectangular samples with a size of 75 × 100 × 115 mm were machined from the as-received material.

The thermomechanical treatment process of the three samples is schematically shown in Figure 1. The samples were first subjected to multiaxial forged deformation at high, medium, and cryogenic temperatures (hereafter labeled HTD, MTD, and CTD, respectively, for simplicity). The multiaxial forging deformation includes three passes, and the forging axis during each pass was changed by 90°. The true logarithmic strain of each pass was 0.2, and the total true logarithmic strain was 0.6 after forging deformation. The initial forging temperatures of HTD, MTD, and CTD were 450, 250, and −180 °C, respectively. The dies for the HTD and MTD samples were heated to 400 °C (Figure 2a), while the dies for the CTD samples were cooled to −180 °C (Figure 2b). The post-forging temperatures of HTD, MTD, and CTD were 400, 200, and −60 °C, respectively. The forged samples were subjected to solid-solution treatment at 475 °C, with a holding time of 4 h. After water quenching at room temperature, the samples were subjected to artificial aging treatment at 120 °C for 4 h, and then at 165 °C for 4 h.

The microstructural evolution was observed through scanning electron microscopy (SEM, Model; TESCAN MIRA4 LMH, Brno, Czech Republic), electron backscatter diffraction (EBSD, Model; OXFORD X-Max, Abingdon, UK), X-ray diffraction (XRD, Model; X’Pert PRO, Almelo, Netherlands), and transmission electron microscopy (TEM, Model; Talos F200X, Hillsboro, OR, USA). The specimens for microstructure observation were taken from the center region of the L–H plane, and the specimens for SEM observation were mechanically ground with #180, #1000, and #2000 sandpaper. For TEM observation, the thickness of the samples was reduced to 60–80 μm with the aid of double-jet polishing. The solution consisted of nitric acid and methanol in a ratio of 3:7. The samples observed by EBSD were prepared by electropolishing in the same nitric acid and methanol solution at −15 °C. The X-ray diffraction (XRD) observation was tested using an X’Pert PRO X-ray diffractometer with a Cu Kα radiation source of 1.5406 Å. Three dog-bone-shaped specimens under each deformation condition were taken from the center of the forging samples to measure the mechanical properties (Figure 3a). The picture and dimensions of the tensile sample are shown in Figure 3b. A universal tensile tester (Model; Instron 3369, Boston, MA, USA) was used to obtain the strength and plasticity of the 7085 alloys with a speed of 2 mm/min.

## 3. Results

### 3.1. Microstructure Analysis after Multiaxial Forging

#### 3.1.1. Second-Phase Particles

Figure 4a shows the morphology and distribution of the second-phase particles for the as-received 7085 alloy. Two kinds of second phase particles were observed in the samples: particles higher in quantity and smaller in size dispersed in the interior of the aluminum matrix, and coarser particles agglomerated inside the matrix. According to the energy dispersive spectroscopy (EDS) analysis, the second-phase particles were Cu, Mg, and the Zn-rich phase. The particle distribution in the HTD sample showed that the dispersed particles in the initial material dissolved into the Al matrix after hot deformation, while a large number of agglomerated particles still existed in the sample (Figure 4b). The high temperatures provided the energy needed for the diffusion of atoms and thus the dissolution of dispersed particles. When the deformation temperature decreased to 250 °C, only some of the dispersed particles were dissolved, and many dispersed and coarse particles were observed in the sample (Figure 4c). As the temperature further decreased to −180 °C, the energy for atomic diffusion decreased. Thus, the number of dispersed particles was similar to that of the initial material (Figure 4d). Interestingly, fragmented particles were observed in the matrix, which was caused by the high shear deformation force at cryogenic temperatures. The above results show that high temperatures can promote the dissolution of particles, while low temperatures cause the fragmentation of particles.

#### 3.1.2. Dislocation Density

XRD tests were carried out to evaluate the dislocation density of the 7085 alloys after forging deformation (Figure 5a). The peaks of the (111), (200), (220), and (311) planes were observed. As shown in Figure 5b, the full width at half maximum (FWHM) values at different peaks gradually increased with decreasing temperature, which indicated higher lattice distortion in the low-temperature deformed samples. Notably, the FWHM values of the CTD sample were far higher than those of the HTD and MTD samples, as cryogenic temperatures can effectively suppress the recovery of dislocation during the deformation process.

The dislocation density was quantitatively tested by the Williamson–Hall method, which can be expressed as follows:(1)$$\frac{\beta \cos\theta }{\lambda }=\frac{1}{D_{v}}+2e\left(\frac{2\sin\theta }{\lambda }\right)$$
where λ is the wavelength of Cu Kα radiation; θ is the diffraction angle; and *e* and *D*_v_ respectively represent the microstrain and subgrain sizes. Thus, *e* and *D*_v_ can be calculated by fitting (2 sin θ/λ) and (β cos θ/λ). Then, the dislocation density (*ρ*) can be described as Equation (2):(2)$$\rho =\frac{2\sqrt{3}e}{db}$$
where *b* is the Burgers vector for Al (0.286 nm). Using Equations (1) and (2), the *ρ* values for the HTD, MTD, and CTD samples were calculated as 1.8 × 10^12^, 3.8 × 10^12^, and 2.0 × 10^13^ m^−2^, respectively.

#### 3.1.3. Dislocation Morphology

TEM observations were conducted to further study the dislocation morphology of the deformed samples. As shown in Figure 6a, few dislocations were observed in the HTD sample, which was attributed to the high recovery and recrystallization ability when deformed at high temperature. Some unbroken coarse particles were found in the sample. Dislocation lines and dislocation tangles were observed in the grains of the MTD sample (Figure 6b). Subgrains with an average size of 500 nm were found in the deformation zones, and the dislocation density of the interior grain was very low. The subgrains had typical recovery characteristics, which were transformed by dislocation cells driven by thermal activation. As the deformation temperature decreased to −180 °C, dynamic recovery was significantly suppressed, and many dislocation tangles and dislocation cells were observed in the grains (Figure 6c). Further decreasing the deformation temperature caused more intense dislocation tangles and poorly defined grain boundaries, with an average size of 200 nm observed. The TEM results revealed that finer substructures were produced when the alloy was subjected to forging deformation at a lower temperature.

### 3.2. Microstructure Analysis after T6 Treatment

#### 3.2.1. Second-Phase Particles

Figure 7a–c shows the SEM images of the 7085 alloys after solid solution and artificial aging treatment, and the corresponding statistical comparison of the coarse particles is shown in Figure 7d,e. The dispersed particles inside the forged samples were almost dissolved in the Al matrix after heat treatment, while coarse particles were still observed in the samples. This is because the dissolution ability of the smaller dispersed particles was higher compared to the coarse particles. The proportions of coarse second-phase particles for the HTD, MTD, and CTD samples were 0.27%, 0.23%, and 0.2%, respectively, which indicated that less deformation promoted the dissolution of particles. Meanwhile, the coarse particles were agglomerated in the HTD samples, with an average particle size of 8.2 μm (Figure 7d). The particle size of the MTD sample (8.1 μm) was close to that of the HTD sample (Figure 7e). As the deformation temperature decreased to −180 °C, however, the average size of the particles decreased significantly to 6.9 μm, and the particles in the sample became more dispersed. Coarse particles had fragmented into finer particles during the cryogenic forging process, increasing their dissolution ability during the subsequent heat treatment process.

#### 3.2.2. Grain Structure

The grain structure and misorientation frequency of samples after heat treatment are shown in Figure 8. The HTD sample exhibited a typical hot deformation grain structure, which contained original coarse and fine grains formed by dynamic recrystallization (Figure 8a). Additionally, there are many low-angle grain boundaries in the original coarse grains. For the MTD sample, many equiaxed grains caused by static recrystallization were observed, where few low-angle grain boundaries were found (Figure 8b). As indicated in Section 3.1.2 and Section 3.1.3, a large number of dislocations were stored in the grains, when the forging temperature decreased from 450 to 250 °C, which promoted static recrystallization during solid solution treatment. Meanwhile, grains that had not undergone static recrystallization were also observed. When the forging temperature decreased to −180 °C, dynamic recovery was significantly suppressed during the forging process, which increased the stored energy and number of nucleation sites for static recrystallization. Consequently, the fraction of static recrystallized grains was further improved, and the average grain size decreased.

#### 3.2.3. Precipitation Phase

The TEM bright-field images of the 7085 alloys after artificial aging treatment with a beam direction along the <110>_matrix_ are shown in Figure 9. For the 7085 alloys, the main strengthening phase was the η’ phase, and two out of four possible η’ variants, i.e., disc-like and needle-like morphologies, were observed [18,19]. The distribution of the precipitation phase was inhomogeneous for the HTD sample (Figure 9a), where the precipitation-free zone (PFZ) and intense precipitation coexisted (as indicated by the arrows). The needle-like η’ phase was 6 nm in length and 2.5 nm in width, and the average diameter of the disc-like η’ phase was 8.5 nm. Figure 9d presents the precipitate morphology of the HTD sample at the grain boundary. The coarsened η (MgZn_2_) phase was distributed intermittently at the grain boundary, and a PFZ with a width of 50 nm was observed. For the MTD sample, the density of the η’ phase interior grains increased compared to that of the HTD sample, and the length of the η’ phase decreased to 5.5 nm (Figure 9b). The width of the PFZ at the grain boundary also decreased to 25 nm (Figure 9e). For the CTD sample, an intense η’ phase with a more homogeneous distribution was observed (Figure 9c). Meanwhile, the PFZ at the grain boundary decreased significantly but did not completely disappear (Figure 9f). The above results revealed that decreasing the temperature promoted the precipitation of the η’ phase of interior grains and suppressed the formation of the PFZ.

### 3.3. Mechanical Properties and Fracture Surface

Figure 10 shows the mechanical property test results of the 7085 alloy after artificial aging treatment, and the corresponding SEM images of the fracture surface are presented in Figure 11. The yield strength, ultimate tensile strength, and elongation of the as-received sample (AR) were 161 MPa, 245 MPa, and 20.2%, respectively. The HTD sample exhibited a high ultimate tensile strength of 539 MPa and yield strength of 514 MPa, but had a low elongation of failure of 7.4%. The number of dimples on the surface of the fracture was small and there were many cleavage fractures (Figure 11a). In addition, many particles were seen on the bottom of dimples (Figure 11d). Compared with the HTD sample, the MTD sample exhibited a higher elongation of failure of 10.2% and lower ultimate tensile and yield strengths of 533 and 495 MPa, respectively. The number of dimples of the MTD sample increased significantly (Figure 11b), and the depth of the dimples was higher (Figure 11e), signifying their excellent plasticity. The CTD sample exhibited high comprehensive mechanical properties, with an ultimate tensile strength of 535 MPa, a yield strength of 506 MPa, and an elongation of failure of 11.1%. A large number of intergranular fracture features were also observed (Figure 11f).

## 4. Discussion

From the experimental results described in Section 3, it can be concluded that the microstructures (i.e., second-phase particles, grain morphology, and precipitation phase) of the 7085 alloys changed significantly during the thermomechanical treatment process, which further affect the strength and plasticity of the alloy. The mechanism of microstructure evolution and the relationship between the microstructures and mechanical properties are discussed further.

The coarse second-phase particles of the 7085 alloys are mainly composed of Zn, Mg, and Cu, which were fragmented and dissolved in the Al matrix during the deformation and heat treatment process. By decreasing the deformation temperature, the hardness of the coarse particles increased while their plasticity decreased. Meanwhile, the flow stress of the aluminum alloy gradually increased as the deformation temperature and the larger lattice friction stress increased the resolved shear stress [20]. Consequently, the particles fragmented into smaller particles during the multiaxial forging process under low-temperature conditions, especially at cryogenic temperatures. In the subsequent solid solution treatment, the broken particles, which were smaller in size, had a higher dissolution ability due to the larger interface between the particles and the Al matrix [21]. In addition, according to Section 3.1.1 and Section 3.1.2, more dislocations were stored in the cryogenic forged sample, which can provide channels for atom diffusion, and thus promote the further dissolution of particles. As a result, more particles in the CTD sample disappeared in the thermomechanical process.

In regard to the grain structure of the 7085 alloys, dynamic recrystallization occurred during the hot deformation process, and many small grains formed in the sample. Dynamic recrystallization can be divided into continuous recrystallization (CDRX) and discontinuous recrystallization (CDRX), according to its nucleation mechanism. DDRX grains usually form at grain boundaries, while CDRX grains appear in the triangular grain boundary region with transition grain boundaries or within grains [22,23]. The CDRX usually occurred at a higher temperature or a lower strain rate. In this work, the hot deformation temperature was 450 °C, which is high for aluminum alloys. Consequently, the dominant recrystallization mechanism of the HTD sample was CDRX. However, the dynamic recrystallization process was not complete, and many coarse grains were retained in the HTD sample. Many subgrain boundaries formed the interior of the coarse grains due to their higher recovery ability at higher temperatures. Compared to the HTD sample, dynamic recrystallization was suppressed in the MTD and CTD samples because of the lower deformation temperature, and many dislocations of tangles, cells, and substructures were stored in the sample after multiaxial forging deformation. In the subsequent solid solution treatment, the dislocations and substructures acted as nucleation sites for static recrystallization. The grain size was mainly determined by the number of nucleation sites, and the CTD sample thus had a smaller grain size. Meanwhile, the distribution of grains through static recrystallization was more homogeneous than through dynamic recrystallization.

The second-phase particle and grain structure can affect precipitation behavior in the artificial aging treatment. By decreasing the deformation temperature, more coarse particles were dissolved into the matrix. This increased the supersaturation of solute atoms after water quenching, which promoted precipitation in the aging treatment [24]. Consequently, a more intense precipitation phase formed in the low-temperature deformed sample. The PFZ can be significantly reduced by decreasing the proportion of particles. The grain distribution also affects the precipitation behavior of the 7085 alloys. For the HTD sample, a large number of subgrain boundaries existed in the grains, which promoted the inhomogeneous distribution of the η’ phase. For MTD and CTD samples, however, the dominant grain boundary was a high-angle boundary due to a higher recrystallization fraction, with fewer low-angle boundaries in the recrystallized grains. As a result, the precipitation distributions in MTD and CTD were more homogeneous.

The relationship between the microstructure and the mechanical properties can be interpreted as follows. The HTD sample is composed of many precipitation phase interior grains, which effectively increased the resistance of the dislocation movement and thus increased the yield and ultimate tensile strengths. Additionally, the low-angle grain boundaries and recrystallized grains also increased the strengthening. The strength values of the HTD sample were consequently higher than those of the MTD and CTD samples. However, many coarse second-phase particles were retained in the sample following heat treatment, providing nucleation sites for dimples during tensile tests, and significantly decreasing its plasticity [25]. The existence of many particles on the fractured surface confirmed the above analysis (Figure 11c). For the MTD sample, many high-angle grain boundaries formed by static recrystallization were observed in the sample. According to the literature [26], the strengthening ability of the recrystallized grains was lower than that of non-recrystallized grains, because few subgrain boundaries existed in the sample. The distribution of the second-phase particles of the MTD sample was more homogeneous, and the number of particles also decreased. As a result, the plasticity of the MTD sample was significantly higher. Further decreasing the temperature to −180 °C caused more particles to dissolve in the Al matrix. An intense precipitation phase formed in the sample, and the PFZ decreased significantly. Although the fraction of high-angle boundaries of the CTD sample was further increased compared to that of the MTD sample, the intense precipitation phase increased its strength. Meanwhile, the plasticity of the CTD sample was satisfactory, with a fracture elongation of 10.7%.

## 5. Conclusions

The effects of deformation temperature on the microstructures (i.e., second phase particles, grain structure, and precipitation phase) and mechanical properties of 7085 alloys were studied. The main conclusions are summarized as follows:

(1)The dispersed particles were dissolved in the Al matrix after hot deformation, while coarse particles were broken into finer particles. Lower deformation temperatures can promote the dissolution of coarse second-phase particles during the heat treatment process by introducing more dislocations. The proportions of coarse particles for HTD, MTD, and CTD were 0.27%, 0.23%, and 0.2%, respectively.(2)The grain structure of the HTD sample was inhomogeneous, with coarse grains and new grains formed by dynamic recrystallization coexisting in the sample. The grain structure of the MTD and CTD samples contained many grains formed by static recrystallization. The grain morphologies of MTD and CTD were more equiaxed compared to that of HTD.(3)Fewer coarse particles in the CTD sample led to the higher supersaturation of solute atoms after quenching, and thus promoted the precipitation of the strengthening phase during artificial aging treatment. Meanwhile, the PFZ was significantly suppressed in the sample where the number of coarse particles was fewer.(4)The refinement of coarse particles, grain size, and precipitation led to the high comprehensive mechanical properties of the CTD sample, with an ultimate tensile strength of 535 MPa, a yield strength of 506 MPa, and a fracture elongation of 11.1%.

## Figures and Tables

**Figure 1 materials-15-06955-f001:**
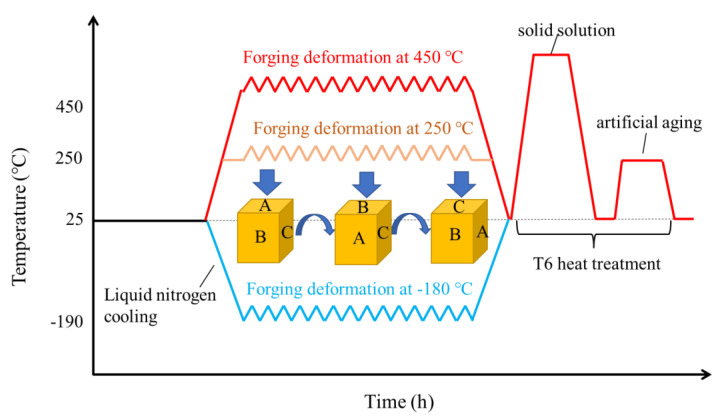
Schematic image of the thermomechanical processing procedures.

**Figure 2 materials-15-06955-f002:**
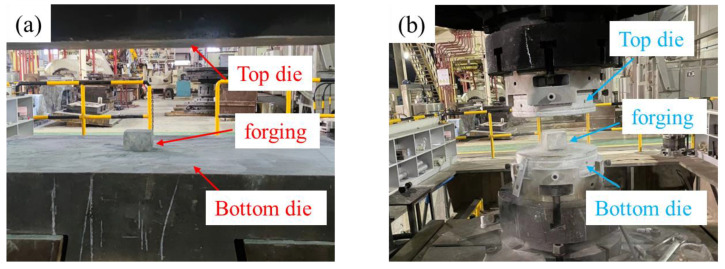
Multiaxial forging deformation setups: (**a**) HTD and MTD; (**b**) CTD.

**Figure 3 materials-15-06955-f003:**
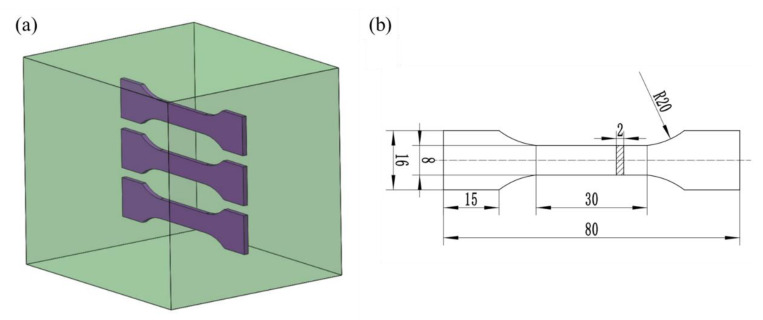
(**a**) Schematic diagram of the samples used in the tensile test; (**b**) dimensions of a tensile sample.

**Figure 4 materials-15-06955-f004:**
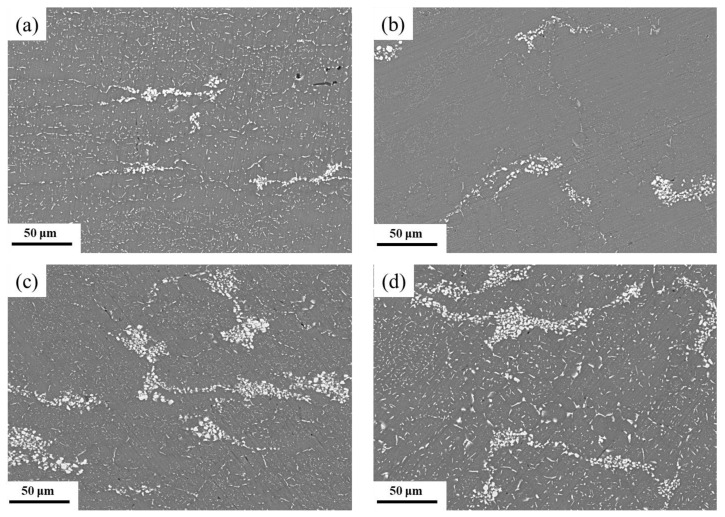
SEM images of the 7085 alloys: (**a**) initial sample without deformation and the sample deformed at (**b**) 450, (**c**) 250, and (**d**) −180 °C.

**Figure 5 materials-15-06955-f005:**
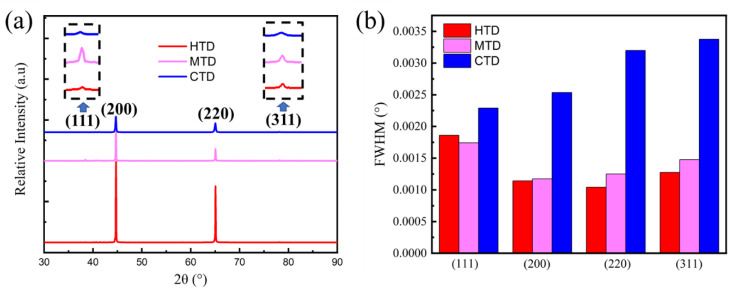
(**a**) XRD patterns and (**b**) FWHM values of the 7085 forged alloys.

**Figure 6 materials-15-06955-f006:**
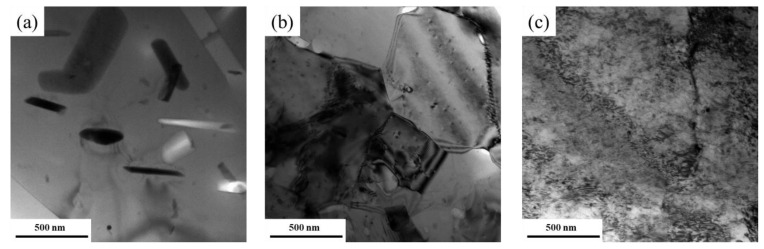
TEM images of the 7085 alloy after forging deformation: (**a**) HTD; (**b**) MTD; (**c**) CTD.

**Figure 7 materials-15-06955-f007:**
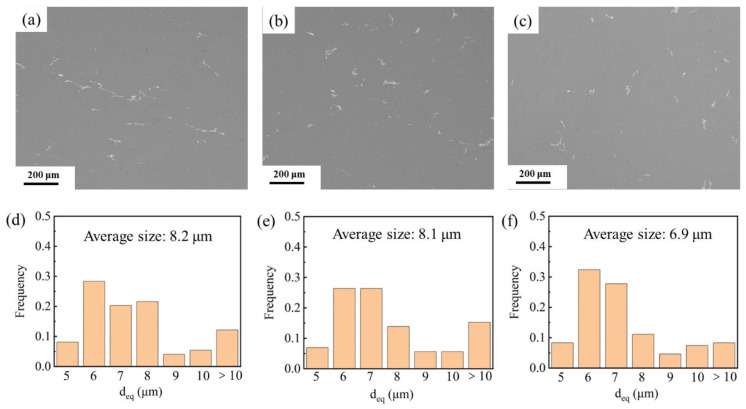
SEM images and particle size distribution of the 7085 alloys after T6 heat treatment: (**a**,**d**) HTD; (**b**,**e**) MTD; (**c**,**f**) CTD.

**Figure 8 materials-15-06955-f008:**
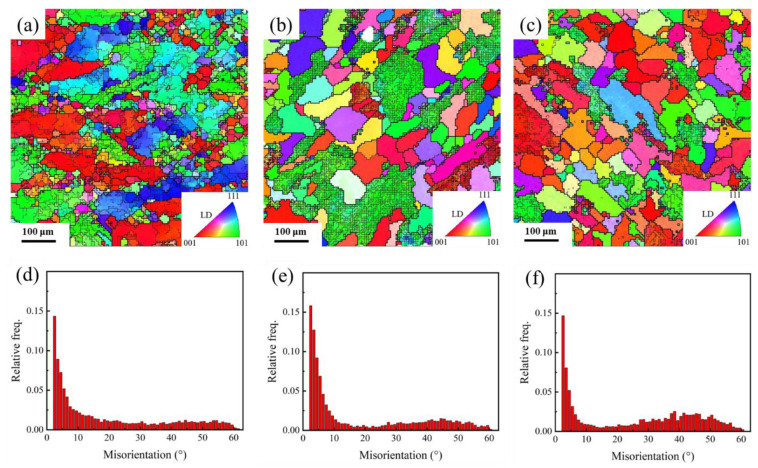
EBSD images and misorientation frequency of 7085 alloys after T6 heat treatment: (**a**,**d**) HTD; (**b**,**e**) MTD; (**c**,**f**) CTD.

**Figure 9 materials-15-06955-f009:**
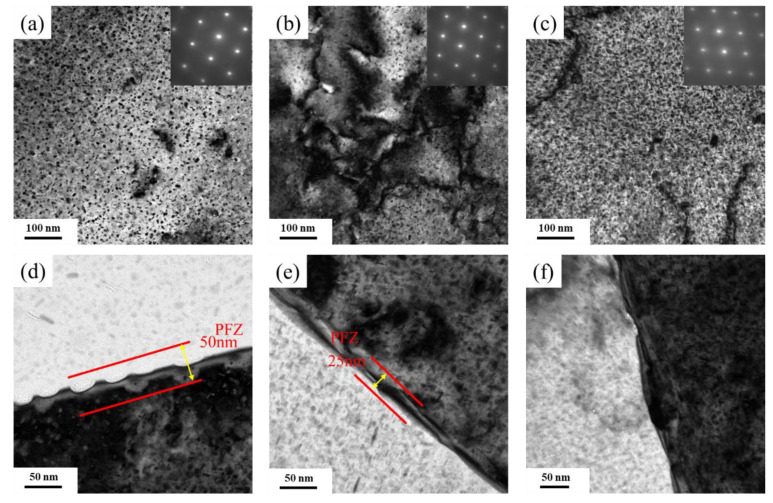
TEM images of the 7085 alloys after T6 heat treatment: (**a**,**d**) HTD; (**b**,**e**) MTD; (**c**,**f**) CTD.

**Figure 10 materials-15-06955-f010:**
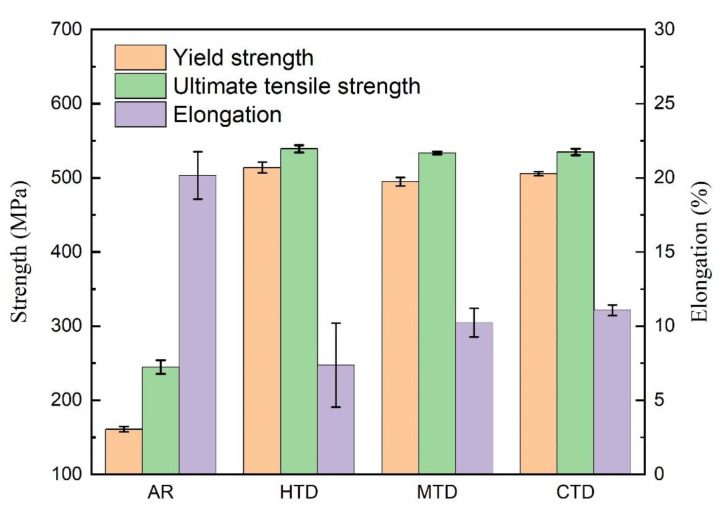
Stress–strain curves of T6 heat treated 7085 alloys under different deformation conditions.

**Figure 11 materials-15-06955-f011:**
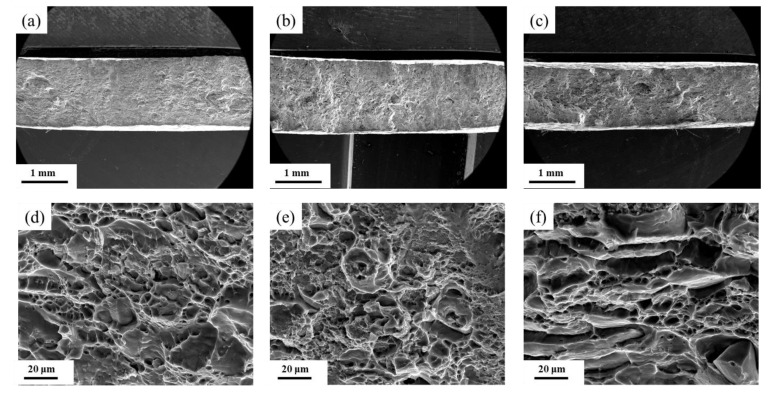
Fracture surface characteristics of 7085 alloys after tension tests: (**a**,**d**) HTD; (**b**,**e**) MTD; (**c**,**f**) CTD.

## Data Availability

Informed consent was obtained from all subjects involved in the study.

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
