# Peer review of "Effects of Deformation at High, Medium, and Cryogenic Temperatures on the Microstructures and Mechanical Properties of Al-Zn-Mg-Cu Alloys"

_materials, 2022, doi:10.3390/ma15196955_

Round 1
Reviewer 1 Report
The manuscript entitled “Effects of deformation at high, medium, and cryogenic temperatures on the microstructures and mechanical properties of Al– Zn–Mg–Cu alloys” has been submitted by authors. Some issues to be addressed which will improve the quality of manuscript. Therefore, I recommend this work could be published after the major revision
1. What is the novelty of this paper?
2. The English composition requires many improvements. The authors should proofread the manuscript carefully to minimize grammatical errors.
3. Check the format of the reference and correct all the errors.
4. All the references mentioned in the paper should be cited in the text or vice-versa.
5. The table and figures heading should be incorporated and discussed in the text.
6. All grammatical errors in the manuscript should be rectified.
7. This research topic is widely studied in the past and a lot of studies are performed. Author please added comparative table for reader clear understanding.

Author Response
On behalf of my co-authors, we thank you very much for giving us an opportunity to revise our manuscript, we appreciate editor and reviewers very much for their positive and constructive comments and suggestions on our manuscript entitled “Effects of deformation at high, medium, and cryogenic temperatures on the microstructures and mechanical properties of Al–Zn–Mg–Cu alloys”. (ID: Materials-1901518). Those comments are very helpful for revising and improving our paper, as well as the important guiding significance to other research. In the file “revised manuscript”, the words marked in red are the portion that the manuscript was amended, and the responds to the reviewers’ comments are as follows (the replies are highlighted in blue).
The manuscript entitled “Effects of deformation at high, medium, and cryogenic temperatures on the microstructures and mechanical properties of Al– Zn–Mg–Cu alloys” has been submitted by authors. Some issues to be addressed which will improve the quality of manuscript. Therefore, I recommend this work could be published after the major revision.
- What is the novelty of this paper?
Response: We apologize for the lack of an adequate description of the innovation of the paper. Al–Zn–Mg–Cu alloys are used to fabricate numerous components because of their lightweight, including the propellant tanks of launch vehicles and armored-vehicle plating. Despite the advantages of these alloys, such light, high-performance, safe alloys require further development; hence Al–Zn–Mg–Cu alloys with superior microstructures are required. In this regard, it is important to understand how various deformation treatments affect the mechanical properties of alloys to improve them for a wider range of applications. In this study, we investigate how deformation temperature affects the second-phase particles, grain structure, and mechanical properties of the 7085 Al alloy, an Al–Zn–Mg–Cu alloy. The coarse second-phase particles and grains were refined by applying cryogenic deformation, followed by solution treatment. We show that coarse particles are fragmented at cryogenic temperature and that higher dislocation densities in the cryogenically deformed samples promote the dissolution of particles during subsequent solution treatment owing to their lower activation energies for atomic diffusion. Further, microstructural refinement increased the ultimate tensile strength, yield strength, and elongation of the Al–Zn–Mg–Cu alloy.
- The English composition requires many improvements. The authors should proofread the manuscript carefully to minimize grammatical errors.
Response: We apologize for the poor expression and grammatical problems. We have carefully corrected the grammar and spelling errors based on your suggestions. In addition, we have asked native English editors to polish and modify the manuscript. The certificate of language editing is shown in the attached image. We hope that the language of our revised manuscript can meet the professional requirements for publication in Materials.
- Check the format of the reference and correct all the errors.
Response: We are sorry for the wrong format of references in the original manuscript. We have corrected the errors in the revised manuscript.
- All the references mentioned in the paper should be cited in the text or vice-versa.
Response: We proofread the manuscript carefully, and all references have been cited in the text.
- The table and figures heading should be incorporated and discussed in the text.
Response: We are sorry for the absence of discussion of the table and figures. We have carefully corrected the manuscript to ensure that all tables are included and cited.
- All grammatical errors in the manuscript should be rectified.
Response: We have corrected the grammatical errors in our revised manuscript. Thank you again for your suggestion.

Reviewer 2 Report
The paper is well written and fit into the scope of Materials. The experiments were also well-planned and executed with adequately presented results and discussion. I believe the paper is acceptable in its current form. However, the review has some few comments and concerns stated below:
1. The author did not clearly show why they chose the selected high and medium temperatures. Specifically, what is the melting point of the material and how far is 450 and 250 Celsius from the melting point?
2. The Abstract did not discuss the mechanical properties of the 450 and 250 Celsius but only presented that of the cryogenic temperature. Moreover, the authors should have conducted similar experiments on the as-received materials so that their results would have served as reference points for the discussion and comparison.
3. Based on the information provided on page 2, in line 80 to 82, it is doubtful that the samples actually attained the stated multiaxial forging temperatures of 450, 250 and -180 Celsius.
4. The impact of the multiaxial forging is also unclear to the reviewer since results were only presented for samples heat-treated samples after forging.
5. In Fig. 3, why did the author choose this direction shown for the dog-bone samples? There is a possibility for anisotropic behavior after forging. Mechanical tests could have been performed at different orientations.
Author Response
Dear editor and reviewers:
On behalf of my co-authors, we thank you very much for giving us an opportunity to revise our manuscript, we appreciate editor and reviewers very much for their positive and constructive comments and suggestions on our manuscript entitled “Effects of deformation at high, medium, and cryogenic temperatures on the microstructures and mechanical properties of Al–Zn–Mg–Cu alloys”. (ID: Materials-1901518). Those comments are very helpful for revising and improving our paper, as well as the important guiding significance to other research. In the file “revised manuscript”, the words marked in red are the portion that the manuscript was amended, and the responds to the reviewers’ comments are as follows (the replies are highlighted in blue).
Reviewer #2:
The paper is well written and fit into the scope of Materials. The experiments were also well-planned and executed with adequately presented results and discussion. I believe the paper is acceptable in its current form. However, the review has some few comments and concerns stated below:
- The author did not clearly show why they chose the selected high and medium temperatures. Specifically, what is the melting point of the material and how far is 450 and 250 Celsius from the melting point?
Response: We are very sorry for the absence of the reasons for the selected temperatures. The melting point of aluminum is 660 ℃, and the overburnt temperature of 7085 alloy is around 490 ℃. The billet in the deformation process can produce deformation heat, and to avoid the billet reaching overburning temperature, the high deformation temperature was selected as 450 ℃. The medium temperature deformation is mainly used to increase flow stress, promote second phase particle breakage and increase internal storage energy.
- The Abstract did not discuss the mechanical properties of the 450 and 250 Celsius but only presented that of the cryogenic temperature. Moreover, the authors should have conducted similar experiments on the as-received materials so that their results would have served as reference points for the discussion and comparison.
Response: We are very sorry for the absence of mechanical properties of samples deformed at 450 and 250 ℃. We have added the information in the Abstract part. Meanwhile, the mechanical properties of as received 7085 alloy were also tested (shown in Fig. 10 in the revised manuscript).
- Based on the information provided on page 2, in line 80 to 82, it is doubtful that the samples actually attained the stated multiaxial forging temperatures of 450, 250 and -180 Celsius.
Response: Thank you for your nice question. The temperature of billets was controlled as follows. First, the samples were heated or cooled to the target temperatures, and the temperatures were precise because the process was monitored by thermocouples. Then the samples were subjected to deformation in an open space. The temperature of the samples deformed at high/medium temperature will decrease because the temperatures were higher than the air temperature. To avoid the phenomenon, the forging dies were heated to 400 ℃. Similarly, the cryogenic forging dies were cooled to -180 ℃. Although we have taken the above measures, the post-forging temperatures of samples were inevitably changed, and the HTD, MTD, and CTD samples after forging deformation were 400, 200, and −60 °C, respectively. Therefore, 450, 250, and -180 ° C in this paper refer to the initial deformation temperature.
- The impact of the multiaxial forging is also unclear to the reviewer since results were only presented for samples heat-treated samples after forging.
Response: In our revised manuscript, we have discussed the influence of deformation temperature on second-phase particle, dislocation density, and substructural morphology in Sections 3.1.1,3.1.2, and 3.1.3, respectively. The microstructures and mechanical properties of heat-treated samples after forging were studied in Section 3.2. Thank you for your suggestion.
- In Fig. 3, why did the author choose this direction shown for the dog-bone samples? There is a possibility for anisotropic behavior after forging. Mechanical tests could have been performed at different orientations.
Response: It is true as the Reviewer suggested that the mechanical properties of sample at different direction are important considering the existence of anisotropic. However, to test the mechanical properties in the specific direction (Fig. 3a) and observe the microstructures, the forged sample has been wire-cut. We are very sorry that the remaining samples are not sufficient to test the mechanical properties of the alloy in the other two directions.
Compared with rolling deformation and extrusion deformation, the sample in this paper adopted the deformation mode of multi-direction forging, so the anisotropy of mechanical properties may be relatively low. In our future research, we will focus on the effect of multidirectional forging on the anisotropy of mechanical properties of alloys. Thank you again for your suggestion.

Reviewer 3 Report
The authors have described the correlation between microstructural and mechanical properties of HTD, MTD, and CTD samples of Al 7085 alloy. The experiments are adequate, the flow is nice and the discussion is in support of the observed facts. The paper is recommended to be published followed by a few minor amendments:
1. In section 3.1.3, the text indicates that fig 6b is the CTD sample (line 170), whereas Fig 6 says that 6b is the MTD sample. Please correct accordingly.
2. Fig 8: Are a, b, c IPF maps or Euler maps? If it is IPF, then include the index for orientation directions for FCC grains.
3. line 234: mention clearly that the PFZ reduces to 25 nm for MTD samples.
4. Fig 9f: Does CTD eliminate the PFZ completely? why? Does it also change the precipitation size?
5. Section 3.3: how many samples were tested per condition? Is it maintaining the ASTM E8 standard?
6. Line 251: it should be MTD sample, where the number of dimples is more.
7. Fig 10: why do MTD and CTD samples show higher elongation than HTD samples when they all have similar UTS and YS? If increased precipitation is the key to this, then CTD should have the largest elongation among them. Can the authors explain this a bit?
8. The inset of Fig 10 shows visible deformation, possibly crack, in the middle of the MTD and CTD samples instead of having a larger fracture to elongation as compared to the HTD sample. why?
9. line 334: PFZ is significantly decreased for the CTD sample, correct the sentence.
Author Response
Dear editor and reviewers:
On behalf of my co-authors, we thank you very much for giving us an opportunity to revise our manuscript, we appreciate editor and reviewers very much for their positive and constructive comments and suggestions on our manuscript entitled “Effects of deformation at high, medium, and cryogenic temperatures on the microstructures and mechanical properties of Al–Zn–Mg–Cu alloys”. (ID: Materials-1901518). Those comments are very helpful for revising and improving our paper, as well as the important guiding significance to other research. In the file “revised manuscript”, the words marked in red are the portion that the manuscript was amended, and the responds to the reviewers’ comments are as follows (the replies are highlighted in blue).
Reviewer #3:
The authors have described the correlation between microstructural and mechanical properties of HTD, MTD, and CTD samples of Al 7085 alloy. The experiments are adequate, the flow is nice and the discussion is in support of the observed facts. The paper is recommended to be published followed by a few minor amendments:
- In section 3.1.3, the text indicates that fig 6b is the CTD sample (line 170), whereas Fig 6 says that 6b is the MTD sample. Please correct accordingly.
Response: We are very sorry for the errors. We have corrected the errors in the revised manuscript.
- Fig 8: Are a, b, c IPF maps or Euler maps? If it is IPF, then include the index for orientation directions for FCC grains.
Response: We are sorry for the absence of information of EBSD images. The EBSD images in Fig. 8 are IPF maps, and the index for orientation directions for FCC grains has been added to our revised manuscript.
- line 234: mention clearly that the PFZ reduces to 25 nm for MTD samples.
Response: When the deformation temperature decreased from 450 ℃ (HTD) to 250 ℃ (MTD), the width of PFZ decreased from 50 nm to 25 nm. This is because more dislocations were stored in interior grains during the low-temperature process. Part of the dislocations was retained in the sample after solid solution treatment, which promoted precipitation in interior grains. Meanwhile, more solute atoms were dissolved into the matrix because of the dissolution of coarse particles. Consequently, the PFZ was decreased.
- Fig 9f: Does CTD eliminate the PFZ completely? why? Does it also change the precipitation size?
Response: The CTD samples only reduced the width of PFZ, but cannot completely eliminate PFZ (shown in Fig. 9c, f). The CTD samples decreased the precipitation size as a result of more nucleation sites during the aging process.
- Section 3.3: how many samples were tested per condition? Is it maintaining the ASTM E8 standard?
Response: Three samples were tested per condition according to the standard of ASTM E8, and error bars have been added to Fig. 10.
- Line 251: it should be MTD sample, where the number of dimples is more.
Response: We are very sorry for the errors. We have corrected the errors in the revised manuscript.
- Fig 10: why do MTD and CTD samples show higher elongation than HTD samples when they all have similar UTS and YS? If increased precipitation is the key to this, then CTD should have the largest elongation among them. Can the authors explain this a bit?
Response: It is a nice question. We think the elongation of forging samples was mainly determined by the grain structures and second phase particles. The HTD sample contained many dynamic recrystallized grains and subgrains, while the MTD and CTD samples contained many static recrystallized grains. The subgrains and fine dynamic recrystallized grains can increase the strengths but decrease the elongation. Meanwhile, more coarse particles were dissolved in the CTD samples. Consequently, the elongation of the CTD sample was the largest.
- The inset of Fig 10 shows visible deformation, possibly crack, in the middle of the MTD and CTD samples instead of having a larger fracture to elongation as compared to the HTD sample. why?
Response: The inset picture of Fig. 10 is the samples after the mechanical property test. The specimen went through three stages during the tensile process, i.e., uniform deformation, neck, and fracture. The MTD and CTD samples contained a larger number of static recrystallized grains, which increased the elongation but decreased the strengths. The visible deformation in MTD and CTD samples was caused by the necking during the tensile deformation process.
- line 334: PFZ is significantly decreased for the CTD sample, correct the sentence.
Response: We are very sorry for the errors. We have corrected the errors in the revised manuscript.

Reviewer 4 Report
The article « Effects of deformation at high, medium, and cryogenic temperatures on the microstructures and mechanical properties of Al–Zn–Mg–Cu alloys» can be published after taking into account the comments. Pay special attention to the reference
In the introduction, write a little more about the scope of these alloys, pay more attention to the relevance of the study.
In Figure 2, visually, the samples look larger than 75 * 100 * 115 mm - does it seem?
It should be noted that 0.2 is the true logarithmic strain
It is necessary to write how the temperature was controlled during deformation? Was deformation heating of the sample observed?
Lines 111 and 311 are missing in Figure 5a. Figure 5b shows their characteristics. Please explain. If there are lines 111 and 311, please change the scale in Figure 5a
It is necessary to index the reflexes in Figure 9 and add this information to the discussion
There are absolutely no articles by non-Chinese authors in the reference
Visually, the percentage of self-citation exceeds 20%
Author Response
Dear editor and reviewers:
On behalf of my co-authors, we thank you very much for giving us an opportunity to revise our manuscript, we appreciate editor and reviewers very much for their positive and constructive comments and suggestions on our manuscript entitled “Effects of deformation at high, medium, and cryogenic temperatures on the microstructures and mechanical properties of Al–Zn–Mg–Cu alloys”. (ID: Materials-1901518). Those comments are very helpful for revising and improving our paper, as well as the important guiding significance to other research. In the file “revised manuscript”, the words marked in red are the portion that the manuscript was amended, and the responds to the reviewers’ comments are as follows (the replies are highlighted in blue).
Reviewer #4:
The article « Effects of deformation at high, medium, and cryogenic temperatures on the microstructures and mechanical properties of Al–Zn–Mg–Cu alloys» can be published after taking into account the comments. Pay special attention to the reference
- In the introduction, write a little more about the scope of these alloys, pay more attention to the relevance of the study.
Response: Thank you for your nice suggestions. We have added references of 7XXX series alloys to the introduction section. The added parts are in red font in our revised manuscript.
- In Figure 2, visually, the samples look larger than 75 * 100 * 115 mm - does it seem?
Response: The size of the initial billet was indeed 75 × 100 × 115 mm. The size of the sample seems to be larger than 75 * 100 * 115mm, which may be caused by the photo angle.
- It should be noted that 0.2 is the true logarithmic strain
Response: We apologize for the incorrect use of professional words, and we have corrected the errors in our revised manuscript.
- It is necessary to write how the temperature was controlled during deformation? Was deformation heating of the sample observed?
Response: Thank you for your nice question. The temperature of billets was controlled as follows. First, the samples were heated or cooled to the target temperatures, and the temperatures were precise because the process was monitored by thermocouples. Then the samples were subjected to deformation in an open space. The temperature of the sample deformed at high/medium temperature will decrease because the temperatures were higher than the air temperature. To avoid the phenomenon, the forging dies were heated to 400 ℃. Similarly, the cryogenic forging dies were cooled to -180 ℃. Although we have taken the above measures, the post-forging temperatures of samples were inevitably changed, and the HTD, MTD, and CTD samples after forging deformation were 400, 200, and −60 °C, respectively. Therefore, 450, 250, and -180 ° C in this paper refer to the initial deformation temperature.
- Lines 111 and 311 are missing in Figure 5a. Figure 5b shows their characteristics. Please explain. If there are lines 111 and 311, please change the scale in Figure 5a
Response: Thank you for your suggestion. We have added the enlarged picture in Fig. 5a in our revised manuscript.
- It is necessary to index the reflexes in Figure 9 and add this information to the discussion
Response: We are sorry we can't understand what you mean. Could you explain it in more detail. Thank you very much.
- There are absolutely no articles by non-Chinese authors in the reference
Response: Thank you for your nice suggestion. We have added references written by foreign authors.
- Visually, the percentage of self-citation exceeds 20%
Response: According to the Reviewer’s suggestion, the percentage of self-citation has decreased to below 20%.

Round 2
Reviewer 1 Report
The author addressed all comments/suggestions very carefully; I recommended accepting this manuscript in its present form.
Reviewer 4 Report
The authors made corrections to the text of the article and answered my comments quite fully. I believe that the article can be published in its current form.
Regarding Figure 9. Figures 9a 9b 9c in the upper right corner show diffraction patterns. It would be useful to index some reflexes and mention in the discussion - if you have the opportunity to make these additions
Good luck!